# Vocalization–whisking coordination and multisensory integration of social signals in rat auditory cortex

**Rajnish P Rao[1], Falk Mielke[1], Evgeny Bobrov[1,2], Michael Brecht[1]\***

[1]Bernstein Center for Computational Neuroscience, Humboldt University of Berlin, Berlin, Germany; [2]Berlin School of Mind and Brain, Humboldt University of Berlin, Berlin, Germany

**Abstract** Social interactions involve multi-modal signaling. Here, we study interacting rats to investigate audio-haptic coordination and multisensory integration in the auditory cortex. We find that facial touch is associated with an increased rate of ultrasonic vocalizations, which are emitted at the whisking rate (~8 Hz) and preferentially initiated in the retraction phase of whisking. In a small subset of auditory cortex regular-spiking neurons, we observed excitatory and heterogeneous responses to ultrasonic vocalizations. Most fast-spiking neurons showed a stronger response to calls. Interestingly, facial touch-induced inhibition in the primary auditory cortex and off-responses after termination of touch were twofold stronger than responses to vocalizations. Further, touch modulated the responsiveness of auditory cortex neurons to ultrasonic vocalizations. In summary, facial touch during social interactions involves precisely orchestrated calling-whisking patterns. While ultrasonic vocalizations elicited a rather weak population response from the regular spikers, the modulation of neuronal responses by facial touch was remarkably strong.

**\*For correspondence:** michael. brecht@bccn-berlin.de

**Competing interests:** The authors declare that no competing interests exist.

**Reviewing editor**: Michael Häusser, University College London, United Kingdom

## Introduction

Rats are highly social animals that display complex behaviors (aggression, dominance, mating, parental care, and play [*Barnett, 1958*]), which involve the use of multi-modal signaling and sensing (*Brecht and Freiwald, 2012*). These social interactions are initially characterized by anogenital sniffing followed by facial contacts which occur at a constant rate (*Wolfe et al., 2011*). Facial contacts have also been shown to be involved in dominance-related behaviors in naturalistic settings (*Blanchard et al., 2001*). Facial contacts consist of extensive whisker-to-whisker and snout-to-snout touch episodes, largely mediated by macrovibrissae (*Wolfe et al., 2011*). The use of macrovibrissae results in a strong activation of the barrel cortex and, importantly, the neurons appear to represent the social context of these interactions by distinct firing rates (*Bobrov et al., 2014*).

Ultrasonic vocalizations (USVs) form another important component of rodent social interactions (*Brudzynski and Pniak, 2002*; *Wright et al., 2010*). Rats produce two distinct classes of USVs, 22 kHz alarm calls and 50 kHz appetitive vocalizations (*McGinnis and Vakulenko, 2003*; *Brudzynski, 2009*), which serve as indicators of negative and positive affective states, respectively (*Knutson et al., 2002*). While the alarm calls are elicited by a range of threats (dominant or aggressive conspecifics, predators, and aversive stimuli [*Brudzynski, 2009*]), the 50-kHz calls are produced in anticipation of/in response to direct social contact (*Blanchard et al., 1993*; *Bialy et al., 2000*; *Brudzynski and Pniak, 2002*), mating (*Bialy et al., 2000*), and homo-specific (rough-and-tumble) or hetero-specific (tickling) play behaviors (*Burgdorf et al., 2008*). Further, playback experiments have demonstrated that 50-kHz vocalizations can induce approach behavior (*Wöhr and Schwarting, 2007*).

**eLife digest** Rats are highly social creatures, preferring to live in large groups within an established hierarchy. Social interactions range from play, mating, and parental care to displays of aggression and dominance and involve the use of odors, touch, and vocal calls. Touch typically takes the form of snout-to-snout contact, while most vocalizations are ultrasonic, with calls of different frequencies used to signal alarm or pleasure.

To date, most studies of rat vocalizations have involved playback of recorded calls to anaesthetized animals, and relatively little is known about how freely moving rats respond to calls. Rao et al. have now addressed this question by recording video footage of rats interacting with other animals or with objects and then using electrodes to record signals in the brains of these rats.

The video footage revealed that rats produce more vocal calls during social interactions than they do during non-social interactions. Moreover, bursts of calls appear to signal the beginning and end of bouts of snout-to-snout contact, suggesting that rodent communication involves the coordinated use of both tactile and vocal cues. Surprisingly, electrode recordings from the part of the brain that responds to sound—the auditory cortex—revealed that most neurons in this region did not respond to ultrasonic calls. However, a type of neuron called a fast-spiking neuron did respond strongly to these calls.

The work of Rao et al. shows that information from multiple senses is directly combined early in the processing of sensory information. Exactly why tactile stimuli should inhibit the auditory cortex is not clear, but there is some evidence that this may increase the rat's sensitivity to sounds. Further experiments are required to test this possibility and to determine how integrating information from multiple senses affects rodent behavior. This will help us to understand how the brain generates coherent social behaviour from signals arriving through distinct sensory channels.

Studies on the neuronal representation of USVs have largely relied on playback experiments, for example, to demonstrate the induction of c-fos expression (*Sadananda et al., 2008*). Reliable and preferential responses to USVs have been reported upon playback in anaesthetized (*Kim and Bao, 2013*) or awake rats (*Carruthers et al., 2013*). However in the awake auditory cortex, a sparse representation of playback sounds has been demonstrated (*Hromádka et al., 2008*). Little is known about the representation of the whole repertoire of conspecific calls in awake, behaving animals. Likewise, there is little information about the integration of multisensory social signals in the auditory cortex, with the notable exception of experience-dependent modulation of auditory responses by natural odors in mice (*Cohen et al., 2011*). To address these issues, we employed the gap paradigm (*Wolfe et al., 2011*; *von Heimendahl et al., 2012*; *Bobrov et al., 2014*), wherein a combination of USVs and facial touch enables the study of multisensory coordination and integration of social signals in the auditory cortex. Specifically, we pose the following questions: (i) How are calls and whisking related on coarse and fine time scales? (ii) How does auditory cortex respond to calls in interacting animals? (iii) How does auditory cortex respond to facial touch? (iv) Are responses to calls in the auditory cortex modulated by touch?

## Results

To study the neuronal representation of multisensory signaling during social interactions in the gap paradigm (*Figure 1A*), we combined low- and high-speed videography with acoustic and neuronal data acquisition. Rats engaged in extensive facial touch that involved whisker-to-whisker (n = 2894, *Figure 1B*, left) and snout-to-snout contacts (n = 2232, *Figure 1B*, right) with social and non-social stimuli (i.e., conspecifics and objects/plastinated rats, respectively). The interactions were spontaneous, frequent (1.76 events/min), and usually sustained (2.86 ± 2.68 s, mean ± SD; *Figure 1—figure supplement 1A*, *Video 1*). A total of 56,525 USVs were identified and classified into categories described earlier for rats in a social context (*Wright et al., 2010*), that is, trill, complex, flat (*Figure 1C*), upward ramp, downward ramp, inverted u, split, and short (*Figure 1—figure supplement 1B*). A few fear calls were also observed but these were restricted to two animals (*Figure 1—figure supplement 1B*). Analysis of call properties revealed characteristic mean frequencies, bandwidths, and call durations (*Table 1*). Intensity measurements from the four microphones were used to assign a source to the

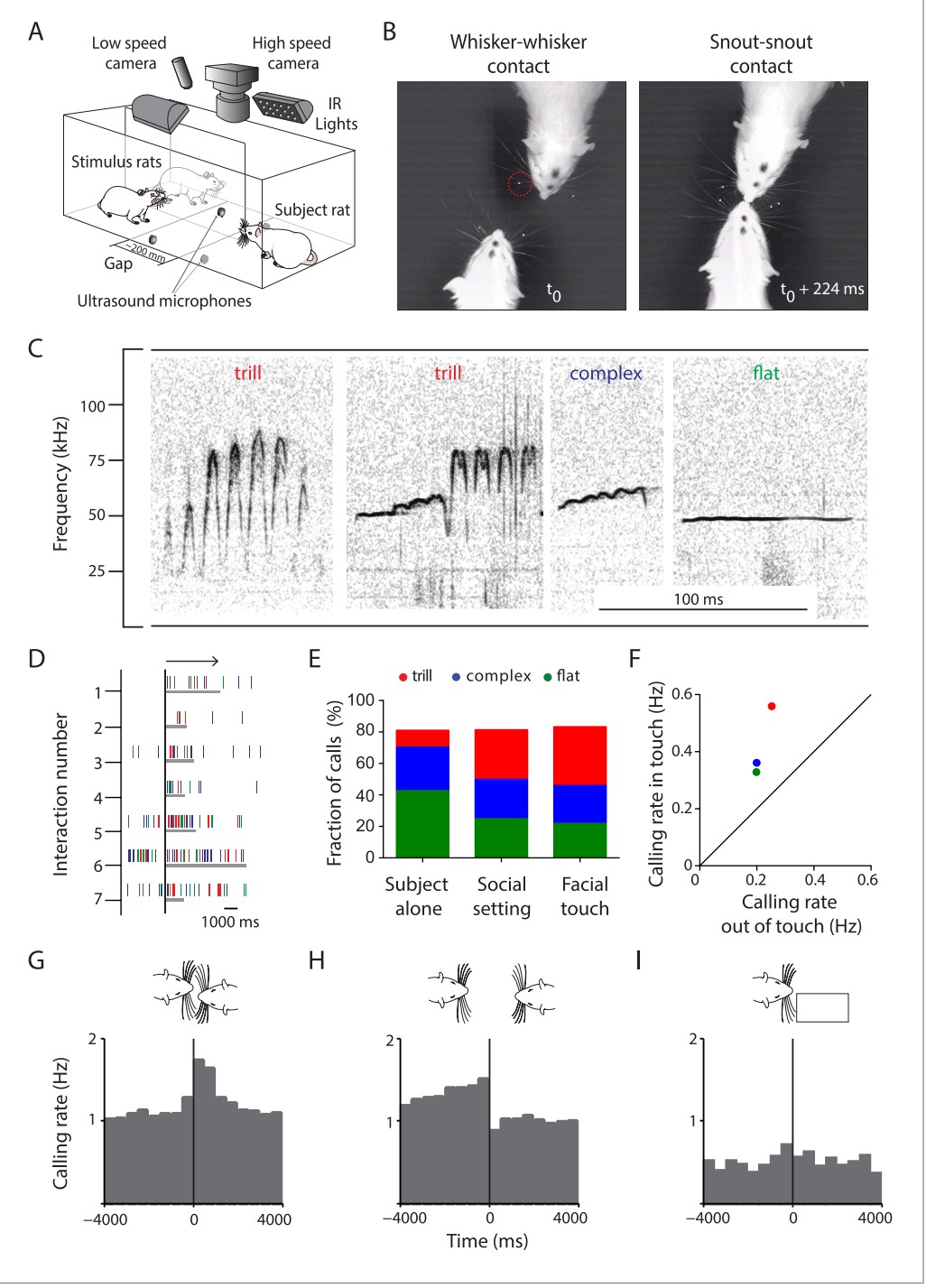

**Figure 1.** Facial touch is associated with increased ultrasonic vocalizations. (**A**) Social interactions between subject and stimulus rats placed across a gap were documented using low- (30 Hz) and high-speed (250 Hz) cameras under infra-red (IR) illumination. USVs were recorded using four ultrasound microphones. (**B**) Facial touch consists of whisker (left) and snout contacts (right). Whisker tracking was facilitated by the use of tags (red circle; $t_0$ = time of first whisker contact). (**C**) Representative spectrograms of three major 50 kHz USV call categories emitted during social interactions, that is, trill, complex, and flat calls. (**D**) Raster plot indicating the relationship between facial touch episodes (gray bars, aligned to start, indicated by arrow) and individual USVs (vertical rasters) in a sample session. (**E**) USVs emitted in three scenarios, by the 'Subject alone', by all interacting partners after introduction of stimulus rats ('Social setting'), and during 'Facial touch' were predominantly trill, complex, and flat calls. *Figure 1. Continued on next page*

*Figure 1. Continued*

The proportion of trills (red) increased in the social contexts while flats (green) showed a reverse trend. Complex calls (blue) remained constant. (**F**) All call types were vocalized at a higher rate during facial touch compared to outside of it (even though stimuli were present). (**G**) Population PSTH aligned to the onset of facial touch indicates an increase in vocalization associated with whisker contact (bin size: 500 ms). (**H**) The end of whisker contact is associated with a sharp decrease in vocalization (bin size: 500 ms). (**I**) Interactions with non-social stimuli (objects/plastinated rats) had lower calling rates and no touch-associated modulation (bin size: 500 ms).

The following figure supplement is available for figure 1:

**Figure supplement 1**. Facial touch is associated with increased ultrasonic vocalizations.

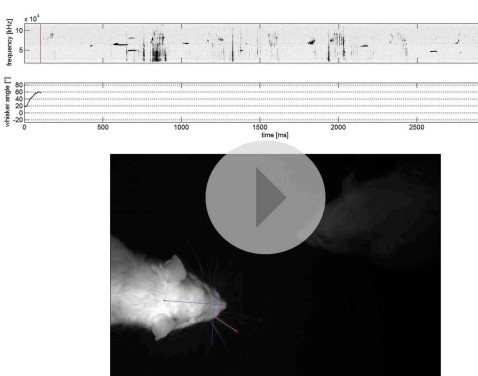

**Video 1**. Temporal coordination of whisking and vocalization. During facial touch in the gap paradigm, rats engage in extensive whisker-to-whisker and snout-to-snout contacts (representative high-speed video slowed down by a factor of 16, bottom panel). In addition, ultrasonic vocalizations produced by the subject rat (left) are shown as spectrograms (slowed down by a factor of 16, top panel). The stimulus animal (right) was anaesthetized to remove confounds of the source of sound. Whiskers of the subject rat were tracked and whisking angles were calculated. A trace of the whisking angle indicates protraction (positive angles) and retraction (negative angles, middle panel). Onsets of individual ultrasonic vocalizations are indicated as vertical rasters in the whisker trace, indicating that vocalization and whisking appear to be temporally coordinated, with a bias towards the retraction phase of the whiskers.

majority of USVs (80%). Analysis of false positive rates estimated that >4 out of 5 calls were correctly assigned to the source (see 'Materials and methods').

## Facial touch is associated with increased ultrasonic vocalization

To study the relationship between calling and social interactions, we aligned the USVs to the onsets of the facial touch episodes (*Figure 1D*). We analyzed three scenarios: (i) when the subject rat was present alone on the setup ('Subject alone'), (ii) after the introduction of stimulus rats during which interactions took place ('Social setting'), and (iii) specifically during the facial touch episodes ('Facial touch'). In all these scenarios, a vast majority of USVs (~80%) were composed of just three call categories, that is, trill, complex, and flat (*Figure 1E*). Interestingly, the proportion of trills was greater in social contexts whereas the proportion of flats was greater when the animals were alone. A similar trend was observed in some of the minor call categories (*Figure 1—figure supplement 1C*). While the baseline calling rate was low when the subject rat was alone (0.13 Hz), a substantial increase (0.80 Hz) occurred in the social setting. Within the facial touch episodes, a further increase in vocalization (1.50 Hz) was observed. Indeed, the calling rate for all call categories increased during facial touch compared to periods outside of it (*Figure 1F*, *Figure 1—figure supplement 1D*), with trills showing the highest increase. Analysis of peri-stimulus time histograms (PSTHs) also revealed an increase in calling associated with the onset (*Figure 1G*) and a sharp decrease with the offset (*Figure 1H*) of facial touch. For interactions with non-social stimuli, the calling rate was lower (0.36 Hz) and there was no increase in calling during touch (*Figure 1I*). On the whole, this increase in calling rate indicates a role for USVs in social communication in our paradigm.

## Calling and whisking are temporally coordinated

Since facial touch was associated with whisker contacts and extensive vocalization, we wondered if whisking and calling were temporally related. To assess this, we tracked precise whisker positions in high-speed videos and mapped the call times onto this information. As shown in the representative example (*Figure 2A*), calls were typically emitted in the retraction phase of the whisking cycle of the call emitter (top trace), whereas there was no systematic relationship of calling to the whisking of the

**Table 1.** Properties of USV categories emitted during social interactions.

| Call type | n | Frequency (kHz) | | | Band width (kHz) | | | Duration (ms) | | |
|---|---|---|---|---|---|---|---|---|---|---|
| | | Mean | ± SEM | 20th, 80th quantiles | Mean | ± SEM | 20th, 80th quantiles | Mean | ± SEM | 20th, 80th quantiles |
| Trill | 16,549 | 63.1 | 0.2 | 51.7, 76.9 | 10.3 | 0.1 | 5.1, 12.6 | 50.9 | 0.2 | 30.2, 67.1 |
| Complex | 13,662 | 54.8 | 0.1 | 48.1, 62.9 | 7.8 | 0.1 | 3.9, 8.7 | 39.0 | 0.2 | 23.8, 49.9 |
| Flat | 13,656 | 49.0 | 0.1 | 38.6, 56.0 | 5.2 | 0.1 | 2.7, 5.3 | 44.1 | 0.3 | 21.7, 56.4 |
| Upward ramp | 1280 | 51.6 | 0.4 | 45.7, 59.8 | 6.5 | 0.2 | 3.5, 8.4 | 26.8 | 0.3 | 15.9, 36.5 |
| Downward ramp | 258 | 48.0 | 1.0 | 31.7, 59.7 | 5.5 | 0.4 | 3.2, 6.8 | 19.9 | 0.8 | 11.6, 25.6 |
| Inverted u | 501 | 55.5 | 0.7 | 45.0, 67.4 | 5.7 | 0.2 | 3.2, 7.2 | 16.4 | 0.5 | 11.0, 18.4 |
| Short | 26 | 49.6 | 3.1 | 36.1, 65.1 | 4.1 | 0.3 | 3.1, 5.4 | 13.7 | 0.9 | 10.9, 15.6 |
| Split | 1072 | 45.2 | 0.5 | 34.0, 57.2 | 7.4 | 0.5 | 3.1, 7.3 | 68.1 | 1.4 | 38.9, 87.8 |
| Fear call | 299 | 28.0 | 0.5 | 22.9, 31.0 | 2.4 | 0.5 | 1.1, 2.3 | 206.5 | 16.6 | 30.2, 353.7 |

interacting partner (bottom trace). Hence, when all whisking traces were aligned and averaged relative to calls, the whisking rhythmicity was preserved in the resulting call-triggered whisking average (*Figure 2B*). We observed that the relationship of call onsets to whisking phase was distributed significantly non-uniformly relative to whisking for call emitter (n = 664, p = 0.0014, *Figure 2C*, top, Hodges–Anje test) but not for interacting partner (n = 705, p = 0.06, *Figure 2C*, bottom). There also appears to be a substantial bias for calls during the retraction phase of the emitter's whisking (381 in retraction vs 283 in protraction, *Figure 2C*, top) unlike for calls during the interacting partner's whisking (352 in retraction vs 353 in protraction, *Figure 2C*, bottom).

We next analyzed the predominant whisking and vocalization rates and observed a peak at ~8 Hz for both by power spectral density analysis (*Figure 2D,E*, left panels). Analysis of the distribution of time intervals between whisking cycles revealed a peak at 112 ms within animals ('auto', *Figure 2D*, middle) but not across animals ('cross', *Figure 2D*, right). This was consistent with previous observations that whisking is not coordinated across animals (*Wolfe et al., 2011*). Similarly, in the analysis of the distribution of time intervals between the start of two subsequent vocalizations of the putative call emitter, a prominent peak was observed at 144 ms ('auto', *Figure 2E*, middle). Such rhythmicity was not obvious when the call onsets across interaction partners were analyzed ('cross', *Figure 2E*, right), suggesting that the coordination of whisking and vocalization is restricted to individuals. This analysis is constrained by the fact that ~1 in 5 calls is incorrectly assigned to a particular call source. Sessions with a single source of USVs (where subject rats interact with plastinated rats/objects) would ideally provide unambiguous evidence as there is only one source of USVs. However, since very few calls are produced in such sessions, we presented subject rats with anaesthetized stimulus rats to elicit extensive vocalization. Analysis of tracked whiskers in the high-speed videos relative to call onset shows that the temporal coordination of calling and whisking is indeed observable (*Video 1*).

## Responses to ultrasonic vocalizations in auditory cortex are excitatory and cell type dependent

To study the neuronal representation of USVs, we performed extracellular tetrode recordings in the auditory cortex (*Figure 3—figure supplement 1A–C*) of awake, behaving rats (four females, four males) while they interacted with conspecifics. Single units were identified based on separation and stability criteria (see 'Materials and methods') and classified as putative regular-spiking (RS) or fast-spiking (FS) neurons (*Figure 3—figure supplement 1D–H*). Analysis of PSTHs triggered to the call onsets revealed the presence of several neurons in the auditory cortex that responded strongly to the calls vocalized within that session (*Figure 3A,C*; *Figure 3—figure supplement 2*).

Strikingly, most auditory cortex neurons showed little or no response to calls while ~10% of RS neurons showed heterogeneous and excitatory responses (*Figure 3A*, *Figure 3—figure supplement 2A–E*). Some RS neurons had a fast onset-associated response (response latency <25 ms of call onset; *Figure 3A*, *Figure 3—figure supplement 2A,B*), while the others showed a more sustained response (*Figure 3—figure supplement 2C–E*). In addition, a few late responders were also observed (response latency

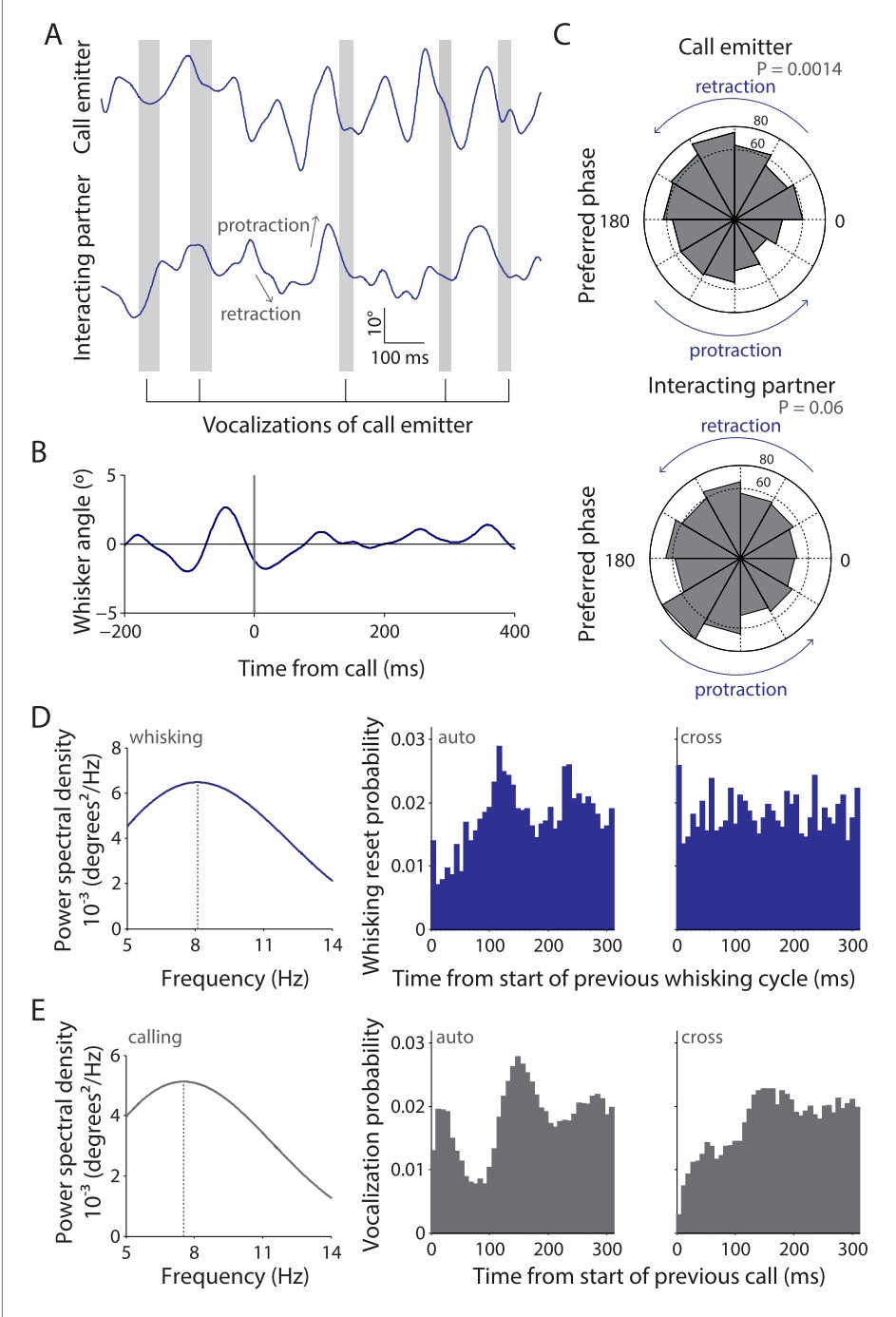

**Figure 2**. Whisking and vocalization are coordinated during facial touch. (**A**) Sample whisker traces of a call emitter (top) when aligned to its own vocalizations (gray bars) indicate a correlation with the retraction phase of the whisker. This was not the case with the whisker trace of the interacting partner (bottom). (**B**) Call-triggered whisking average wherein the average of all whisking traces is aligned with respect to the emitter's call onset. (**C**) Distribution of call onsets were significantly non-uniform (Hodges–Anje test) relative to whisking cycle of emitter itself (top) but not for the interacting partner (bottom). There appears to be a substantial bias for calls during the retraction phase of the emitter's whisking (top) unlike for calls during the interacting partner's whisking (bottom). (**D**) Predominant whisking rates determined by power spectral density of whisker traces during social touch is 8.1 Hz (left). Distribution of time intervals between whisking cycles within one animal (auto) correspondingly shows a peak at ca. 112 ms (middle). Points of maximum protraction were used for binning. Inter-animal (cross) whisking interval distribution suggests

*Figure 2. Continued on next page*

*Figure 2. Continued*

that the interacting animals do not whisk with a fixed phase relationship (right). (**E**) Power spectral density analysis to determine the predominant calling rates revealed a 7.6 Hz predominant frequency component (left). The distribution of time intervals between start of two subsequent vocalizations peaks around 144 ms ('auto', middle). The effect is not as pronounced when triggering to calls that were assigned to a different animal ('cross', right).

---

>50 ms after call onset; *Figure 3—figure supplement 2E*). USVs elicited a small (~5%) but significant increase in the population of RS neurons sampled (*Figure 3B*, left), as reflected in their firing rates (mean response to all calls ± SEM: 9.3 ± 0.82 Hz vs baseline: 8.8 ± 0.77 Hz; n = 172, p = 0.0006, Wilcoxon signed rank test). The median firing rates however were lower (response to calls: 6.06 Hz vs baseline: 5.42 Hz). This would suggest that the overall increase in the mean firing rate was contributed to by a small fraction of neurons within the population. This is in line with the observation that most neurons had little or no response to USVs despite extensive sampling of the auditory cortex across several animals (*Figure 3—figure supplement 1B,C*).

FS neurons on the other hand showed a robust excitatory modulation by USVs (*Figure 3C*, *Figure 3— figure supplement 2F–H*). Most FS neurons had a sustained response to calls (*Figure 3C*, *Figure 3— figure supplement 2F*). In addition, few cells with sharp short latency (*Figure 3—figure supplement 2G*) or delayed (*Figure 3—figure supplement 2H*) responses were also observed. The mean firing rates showed a consistent upregulation (response to calls: 19.11 ± 3.94 Hz vs baseline: 16.92 ± 4.07 Hz, n = 23, p = 0.006, Wilcoxon signed rank test), which is seen in the scatter plots (*Figure 3D*, left).

Analysis of call category specificity revealed that many neurons (RS and FS) responded to more than one of the major call categories (*Figure 3A,C*, *Figure 3—figure supplement 2A,C,D,F–H*). Neurons more strongly modulated by one call category than others were also observed (indicated by asterisks; *Figure 3—figure supplement 2B*). Population level analysis showed that there is a significant modulation of RS neurons by all the major call categories (*Figure 3—figure supplement 3A*, top row). FS neurons however appear to be significantly modulated by trills alone (*Figure 3—figure supplement 3A*, bottom row). The lack of significant modulation by complex and flat calls is probably an artifact caused by the small sample size as most FS neurons show activation in the scatter plots (*Figure 3—figure supplement 3A*, bottom row). We next wanted to check if the population as a whole has a greater preference for one of the call categories. For this, we computed responses to the 'best call' that is, the major call type that elicits the highest modulation. While this scenario biased towards larger effects shows a highly significant modulation by the 'best call' (p < 0.0001, *Figure 3—figure supplement 2B*), there appears to be no overwhelming population of trill-/complex-/flat-preferring neurons.

To compare the responses across cell types and call categories, we computed response indices for each neuron (see 'Materials and methods'). The mean response indices of RS neurons to all calls and the major call categories showed a very small positive modulation (*Figure 3B*, right), with no preference to any specific call category. Mean response indices of FS neurons, on the other hand, showed a robust positive modulation to all calls and to individual call categories (*Figure 3D*, right). Also, the response indices of FS neurons were significantly different from those of RS neurons (p < 0.0001, unpaired Mann–Whitney test).

We also determined that while RS neurons significantly responded to calls from the stimulus animals, this was not the case with the subject animal's own calls (*Figure 3—figure supplement 4A*, top row). However, a pair-wise comparison of each RS neuron's response to own vs stimulus calls was not significantly different (p = 0.85, Wilcoxon signed rank test), suggesting that this was a population level effect. Interestingly, the reverse was true for the FS neurons, which showed a clear preference for own calls but not for the stimulus calls (*Figure 3—figure supplement 4A*, bottom row). Pairwise comparisons of FS neuronal responses showed a significant difference (p = 0.03), suggesting that they could indeed discriminate between own and stimulus calls. We also compared the responses across various auditory cortex sub-fields, that is, auditory cortex dorsal (AuD), primary auditory cortex (Au1), and auditory cortex ventral (AuV). AuD and Au1 RS neurons as well as Au1 FS neurons were significantly modulated by USVs. However, the apparent lack of modulation in AuV RS, AuD, and AuV FS neurons is confounded by the small sample sizes (*Figure 3—figure supplement 4B*).

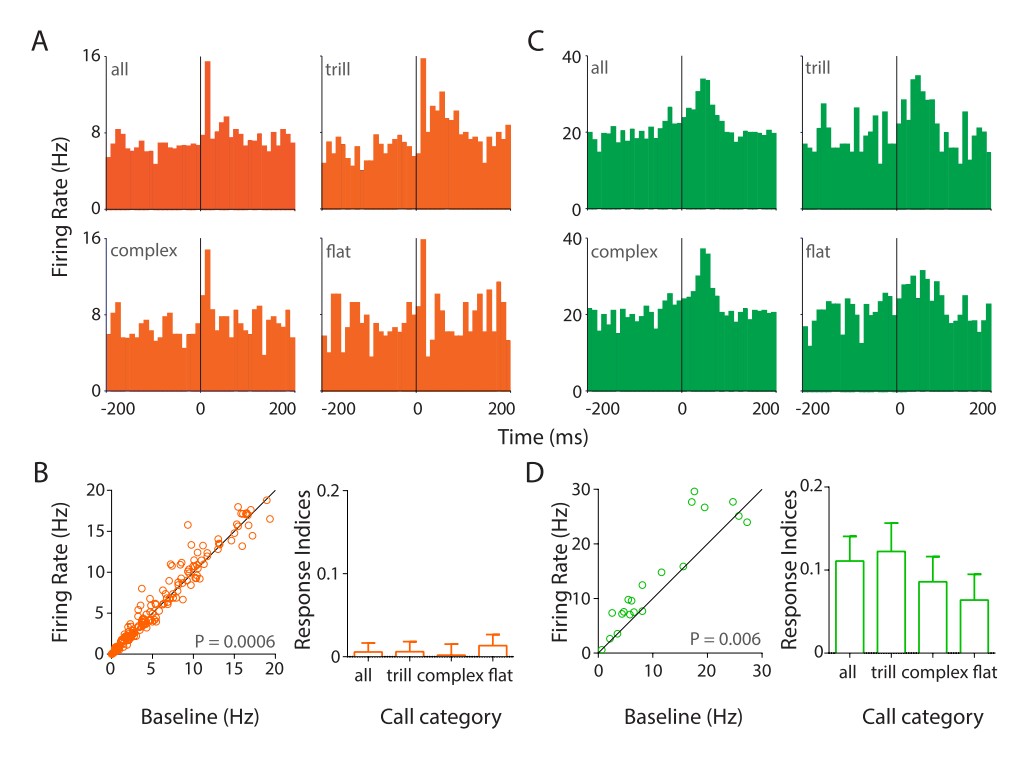

**Figure 3**. Responses to USVs in auditory cortex are excitatory and cell-type dependent. (**A**) Representative PSTHs showing the response of a RS neuron to all USVs (top left) and to individual call categories that is, trill (top right), complex (bottom left), and flat (bottom right) (bin size: 10 ms). (**B**) Population data showing responses of RS neurons plotted as firing rate in response to USVs vs baseline (left, Wilcoxon signed rank test). The population showed a significant modulation in firing rate in response to USVs. The mean response indices ± SEM to either all calls or the major call categories also reveal little or no overall modulation (right). (**C**) Response of a representative FS neuron showing a stronger modulation to all calls and also to complex and trill calls (bin size: 10 ms). (**D**) Almost all FS neurons were also significantly upregulated by USVs (left, Wilcoxon signed rank test) and this was evident in their mean response indices ± SEM to all calls and to the various call categories (right).

The following figure supplements are available for figure 3:

**Figure supplement 1**. Locations of recording sites in the auditory cortex and cell-type classification.

**Figure supplement 2**. Responses of auditory cortex neurons to ultrasonic vocalizations are heterogeneous.

**Figure supplement 3**. Population responses of auditory cortex neurons to different USV call categories.

**Figure supplement 4**. Population responses to own vs stimulus calls and in various sub-regions of the auditory cortex.

**Figure supplement 5**. Auditory cortex neurons do not show any locking to the phase of whisking.

Previously, having observed a correlation between whisking and calling, we next tested if the neuronal firing rate is correlated with the whisker position. Towards this, whisker tracking was performed on high-speed videos to determine whisker angle and phase. Spike time stamps aligned to this information was used to generate polar plots and compute Rayleigh vector lengths for each neuron which showed >10 spikes each in touch episodes. The phase locking was also tested with a shuffling test that is insensitive to the spike count (see 'Materials and methods'). At the population level, there was no strong preference to the whisking phase (*Figure 3—figure supplement 5A*). A few cells that appeared to have a strong locking to the retraction phase were also strong responders to USVs, suggesting that this phase locking is an artifact from locking of whisking and vocalization.

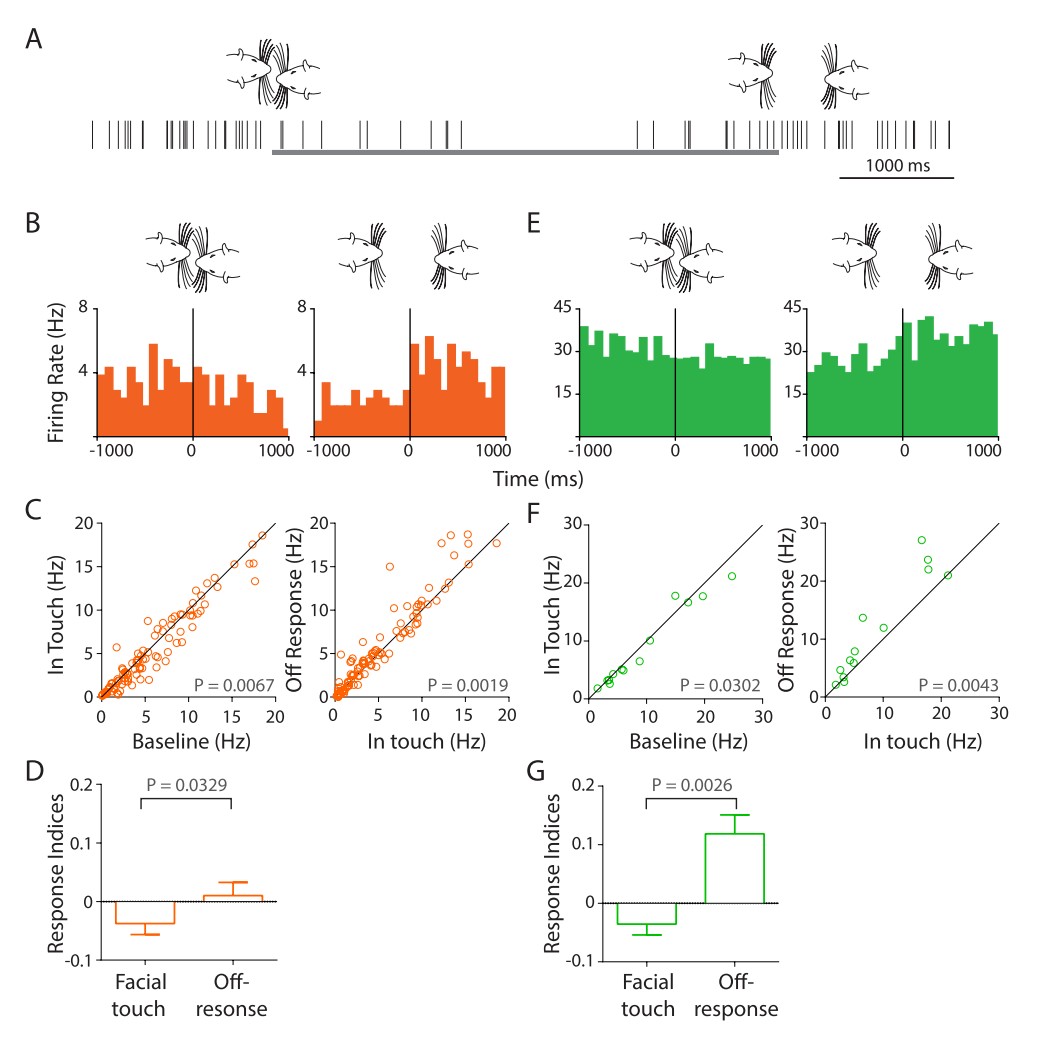

**Figure 4**. Facial touch evokes inhibition and off-responses in primary auditory cortex (Au1). (**A**) Schematic showing the spiking activity (rasters) of a representative Au1 RS neuron aligned to a facial touch episode (gray bar). Facial touch onset results in an inhibition in firing rate and is associated with an increase upon the end of touch. (**B**) Representative PSTHs of a RS neuron triggered to onset (left) and offset (right) of facial touch demonstrating the inhibitory effect during touch and an off-response at the end of touch (bin size: 77 ms). (**C**) Population response of RS neurons plotted as firing rate after touch onset vs baseline (left) shows a significant inhibition due to facial touch. However, the off-response (in a 200 ms window after the end of facial touch) led to an increase in firing rate which was significantly higher than in touch (right, Wilcoxon signed rank test). (**D**) Mean response indices ± SEM of RS neurons during facial touch and off-response were significantly different (Wilcoxon signed rank test). (**E**) Representative PSTHs of a FS neuron triggered to facial touch onset (left) and offset (right) demonstrating a similar inhibitory effect due to touch and the off-response at the end of touch (bin size: 77 ms). (**F**) Population response of FS neurons showing a significant inhibition (Wilcoxon signed rank test) due to touch (left) that was released at the end of touch (right). (**G**) Mean response indices ± SEM of FS neurons were also significantly different (Wilcoxon signed rank test) between facial touch and off-response windows.

The following figure supplement is available for figure 4:

**Figure supplement 1**. Facial touch does not elicit inhibition in AuD and AuV; whisker/snout touch does not lead to measurable changes in sound intensities.

## Facial touch evokes inhibition and off-responses in the primary auditory cortex

Unexpectedly, we observed that facial touch resulted in an inhibition of several RS (*Figure 4A–D*) and FS (*Figure 4E–G*) neurons. Exemplary spike raster (*Figure 4A*) and PSTHs of RS (*Figure 4B*) and FS

(*Figure 4E*) neurons triggered to the onset (left) and end (right) of facial touch show the inhibition elicited due to touch and an off-response, which usually occurred within a 200-ms window at the end of touch. A substantial fraction of neurons in our dataset showed this distinct off-response (~19%). This phenomenon appears to occur in Au1 (*Figure 4C*) but not in the other auditory cortex sub-regions (*Figure 4—figure supplement 1A,B*).

In Au1 RS neurons, the firing rates reduced from 8.18 ± 1.03 Hz to 7.82 ± 1.04 Hz during the touch episode (n = 105, p = 0.0067, Wilcoxon signed rank test, *Figure 4C*, left). Compared to touch, the firing rate in the off-response window was significantly higher at 8.49 ± 1.1 Hz (n = 105, p = 0.0019, *Figure 4C*, right). Unexpectedly, this ~10% increase in the population response was about twice as large as the excitatory responses elicited by USVs in RS cells. Similarly, for Au1 FS neurons, the baseline firing rates decreased from 14.6 ± 4.18 Hz to 13.82 ± 4.03 Hz due to touch (n = 15, p = 0.0302, *Figure 4F*, left). The firing rate during the off-response window was higher at 17.03 ± 4.71 Hz when compared to the touch episode (p = 0.0043, *Figure 4F*, right). The off-response to facial touch is remarkable as it was the largest population response we observed in the auditory cortex. Mean response indices for Au1 RS neurons (*Figure 4D*) showed that the overall inhibition due to touch was reversed after the touch episode (p = 0.0329). A similar inhibition at touch onset and reversal at the end of touch were also evident in Au1 FS neurons (p = 0.0026, *Figure 4G*).

It could be hypothesized that these responses in Au1 are brought about not by the touch itself but by the sounds produced due to touch. To test this, we measured the sound intensities triggered to whisker and snout touch and did not observe any increase in intensity. USVs lead to a large increase as expected (*Figure 4—figure supplement 1C*).

## Responses to calls are modulated by facial touch

Given the robust inhibition observed in Au1, we next asked if facial touch modulates the responses of Au1 neurons to USVs. Indeed, we observed that in RS neurons, the response to calls was modulated by touch (*Figure 5A*). The population response is also evident by an increased modulation by USVs in touch (greater scatter, *Figure 5B*, right) as compared to neurons responding to calls out of touch (*Figure 5B* left). Pairwise comparison of each neuron's response to USVs in (right) and out (left) of touch revealed a highly significant differential modulation (p = 0.0038, Wilcoxon signed rank test, *Figure 5B*). This was also evident in the frequency distribution histograms of the response indices. While response indices of RS neurons to calls out of touch were largely centered around zero indicating little or no modulation (*Figure 5C*, left), they were more spread out during touch (*Figure 5C*, right) indicating significantly greater modulation (p = 0.0072, Kolmogorov–Smirnov test). However, in FS neurons, responsiveness to calls appeared to be less strongly modulated by touch at the individual (*Figure 5D*) and population levels (p = 0.3303, *Figure 5E*). Correspondingly, the spread of activation indices did not differ in and out of touch (p = 0.3752, *Figure 5F*).

The apparent increased modulation in responsiveness to calls during touch could be due to a sampling artifact as only a fraction of calls (19.8%) occur during touch. To rule out this potential confound, we applied bootstrapping analysis (see 'Materials and methods') to compute the mean response indices for our data set and plotted the frequency distribution histograms as before. These bootstrapped data show that neuronal responses in RS (but not FS) neurons were strongly modulated during touch, independent of the number of samples (*Figure 5—figure supplement 1A,B*). Another possible reason for this greater modulation could be the higher effective intensity of vocalizations in touch when the animals were closely juxtaposed. This would suggest that the response of RS neurons would be stronger to stimulus calls in touch while the response to the subject's own calls would not show much modulation (under the assumption that the subject rat vocalizes and perceives its own calls at roughly the same intensities). There, however, does not appear to be any greater responsiveness to stimulus calls during touch. This would suggest that the intensity differences are probably not responsible for the higher modulation during touch (*Figure 5—figure supplement 1C,D*). Taken together, these results suggest that the responsiveness to USVs in the auditory cortex is modulated by facial touch.

## Discussion

### Multi-modal social signaling

Social interactions consist of complex behaviors which employ a range of multi-modal signaling and sensing (*Brecht and Freiwald, 2012*). Social transmission of food preference which relies on the

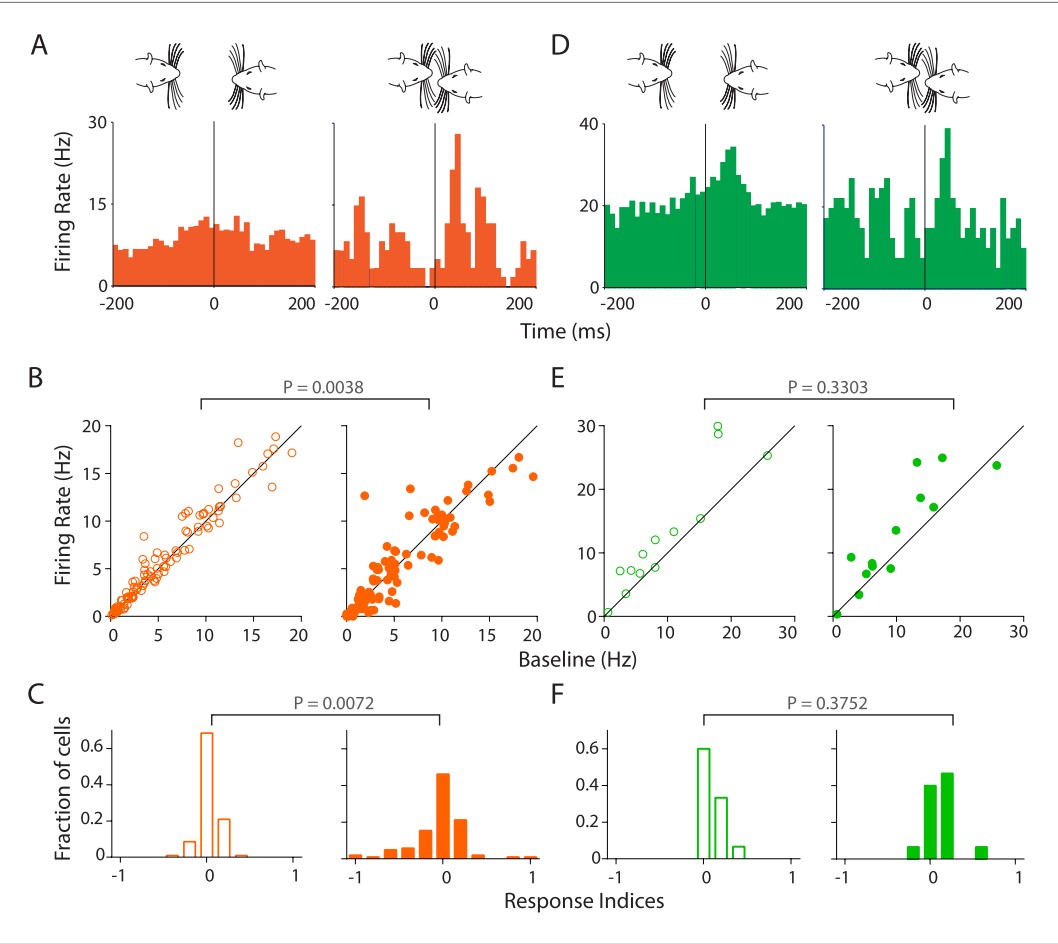

**Figure 5**. Stronger and variable modulation of responses to vocalizations by facial touch. (**A**) Representative PSTHs of a RS neuron showing a stronger response to vocalizations during facial touch (right) compared to out of touch (left, bin size: 10 ms). (**B**) Population response of RS neurons also demonstrated an increased modulation to vocalizations (both excitation and inhibition) during touch (right) compared to outside of it (left), which were significantly different when subjected to a pair-wise comparison (Wilcoxon signed rank test). (**C**) Distribution of response indices showed that a large fraction of RS neurons were not modulated by vocalizations when out of touch (left) while a significant amount of modulation occurred during touch (right, Kolmogorov–Smirnov test). (**D**) Representative PSTHs of a FS neuron showing a higher response to vocalizations during touch (right) compared to the response when out of touch (left, bin size: 10 ms). (**E**) Population response of FS neurons however did not show any significant difference (Wilcoxon signed rank test) in modulation to vocalizations either in (right) or out (left) of touch. (**F**) Distribution of response indices in FS neurons also did not show any significant differences in modulation (Kolmogorov–Smirnov test) either in (right) or out (left) of touch.

The following figure supplement is available for figure 5:

**Figure supplement 1**. Sampling bias does not account for modulation of responsiveness to calls during touch; calls both from own and stimulus animals lead to increased modulation in primary auditory cortex during touch.

combined use of olfactory and gustatory cues (*Galef and Wigmore, 1983*) is probably the most extensively studied example of this. In addition, the use of unimodal signals including vision (ear wiggling [*Erskine, 1989*]), smell (cheek gland pheromones, [*Kannan and Archunan, 2001*], and somatosensation [*Blanchard et al., 2001*]) has also been reported. However, little is known about the multisensory integration of these signals, which would play a critical role in facilitating social interactions. In this inquiry, we study interacting rats to investigate audio-haptic coordination and multisensory integration in the auditory cortex. We demonstrate that facial touch during social interactions is associated with an increased production of USVs. We observe a temporal coordination of vocalization and whisking,

with calls being associated with the retraction of whiskers. USVs elicited excitatory responses in a small fraction of RS neurons in the auditory cortex, whereas almost all FS neurons showed a strong activation. Facial touch however resulted in a robust inhibition of the primary auditory cortex. Moreover, we observed a remarkable off-response at the end of the touch episode, possibly reflecting a release of touch-induced inhibition, which was surprisingly the largest response modulation observed in our study. Finally, it appears that the response of auditory cortex neurons to USVs is modulated by facial touch.

## Role of USVs in social signaling

Previous studies from our laboratory have used the gap paradigm to study social interactions and have demonstrated that spontaneously interacting rats use their whiskers for extensive facial interactions (*Wolfe et al., 2011*; *von Heimendahl et al., 2012*). Interestingly, the social context (sex and sexual status) of these interactions appears to be represented in the barrel cortex neurons (*Bobrov et al., 2014*). A little explored component of these interactions has been USVs, which are known to contribute to species-specific social signaling (*Brudzynski and Pniak, 2002*; *Brudzynski, 2013*). A reduction in social interaction and ultrasonic communication has also been reported in a mouse model of monogenic heritable autism (*Jamain et al., 2008*). Indeed, using the gap paradigm, we observed an increase in the rate of vocalizations during the presentation of social (but not non-social) stimuli. Interestingly, the calling rate also showed a sharp decrease at the end of interactions.

We classified the USVs into specific call types largely based on an earlier description (*Wright et al., 2010*). We observe that a vast majority of calls in our paradigm belong to one of the three categories that is, trill, complex, and flat. Further, the proportion of trills and flats appear to be modulated by the presence of conspecifics. These results are in line with the finding that there is a greater prevalence of flats in singly tested rats whereas trill vocalization is increased in pair-tested rats (*Wright et al., 2010*). A role for trills in social contexts has also been reported during play behavior (*Schwarting et al., 2007*). Taken together, these results would suggest that specific call categories could play a role during specific components of social interactions. This idea is supported by a recent study which reports that male mice emit distinct vocalizations when females leave the environment (*Yang et al., 2013*).

## Temporal coordination of vocalization and whisking

Since facial touch involves the extensive use of whiskers and USVs, we employed high-speed videography to track the position of the whiskers with high temporal resolution and mapped the vocalization data onto this information. We observed a locking of calling to whisking, with calls being produced during the retraction phase of the whisking cycle. The retraction phase has been correlated to the exhalation phase of breathing (*Moore et al., 2013*), which would correspond to call production. Previous studies have shown that in addition to a correlation of respiration and USV production (*Roberts, 1972*; *Riede, 2011*), sniffing and whisking are also tightly correlated (*Welker, 1964*; *Ranade et al., 2013*). The neural substrate of such coordination is thought to lie in the joint architecture of pattern generators for whisking and breathing (*Moore et al., 2013*). Further, our results on the ~8 Hz vocalization rate are in line with an earlier report that there is a selective increase in the representation of sounds repeated at an ethological rate (*Kim and Bao, 2009*). Our findings on the temporally precise audio-haptic coordination in social signaling are reminiscent of much earlier findings on the multisensory orchestration of sensory acquisition (*Welker, 1971*).

## Responses of regular-spiking neurons to ultrasonic vocalizations

RS neurons in the auditory cortex largely had little or no response to USVs. A very small fraction (~10%) however showed robust responses to calls. When we did not observe more numerous responses to USVs, we initially wondered if the freely interacting animals simply did not hear many of the USVs. However, this explanation can be ruled out as: (i) responses to both own and stimulus were the same even though the subject rats were obviously always close to their own calls and (ii) FS cells responded robustly and significantly to calls. To our knowledge, this is the first report of neuronal responses to conspecific calls in awake behaving rats. Previous reports on the neuronal encoding of USVs in the rat auditory cortex have employed playback experiments in anesthetized preparations (*Kim and Bao, 2013*) or awake rats (*Carruthers et al., 2013*). While these studies report reliable population responses to vocalizations, our data are very similar to another study that shows that there is a sparse representation of sounds in the unanaesthetized auditory cortex (*Hromádka et al., 2008*). Using a method that

is not biased to neuronal activity (cell-attached recordings), the authors report that <5% of neurons responded robustly to sounds at any instant. Further, they also report that narrow-spiking inter-neurons are highly responsive.

Of the strongly modulated neurons, we observe a range of responses: most cells displayed early (<25 ms after call onset) increases in firing rates, while a few late responders (>50 ms after call onset) were also observed. Responses tended to be sharp or sustained, with a few cells showing robust modulation by one or more call categories. At the population level, the major call categories evoked strong activation in the RS cell population whereas FS neurons appear to be modulated only by trills. Also, there does not appear to be any population level preference for one of these call types with relation to their best response.

Interestingly, RS neurons seem to prefer the stimulus calls whereas FS neurons were strongly modulated by the animal's own calls. However, a cell-wise analysis of response to own vs stimulus calls does not show any difference, suggesting that this phenomenon is brought out at the population level. The role of FS neurons in blanking out the animal's own calls and facilitating increased response to stimulus calls would be a tempting speculation. There also appears to be auditory cortex sub-region-wise differences, in that AuD and Au1 respond significantly to calls. Despite the temporal coordination of whisking and calling, we do not observe any strong preference of the auditory cortex to the whisking phase at the population level. Rare examples of neurons strongly locked to retraction could be artifacts from locking of whisking and vocalization.

## Response of auditory cortex to facial touch

We next analyzed the response of auditory cortex neurons to facial touch and observed an unexpected inhibition. This was particularly evident as an off-response; wherein at the end of the touch episode, a burst of firing was observed in the PSTHs indicating a possible release from inhibition. This was observed both in RS and FS neurons, but only in those from Au1. These observations are potentially confounded by the fact that whisker/snout contacts could in themselves produce sounds. Indeed, high sensitivity to broadband stimuli has been reported in the cat auditory cortex (*Bar-Yosef and Nelken, 2007*). Sound intensity measurements were performed but no change was detected due to whisker movement and contacts, at least at the microphones. Both social and non-social touch induce inhibition, suggesting that the modulation we observe is likely to be tactile in nature.

An anatomical basis for these observations would lie in the direct connectivity between somatosensory and auditory cortices (*Budinger et al., 2006*). Even prior to the cortical processing, multisensory integration of sound and touch has been shown to occur in the dorsal cochlear nucleus (*Young et al., 1995*; *Kanold and Young, 2001*; *Shore, 2005*). It has also been suggested that a possible role for touch in the auditory areas is to enhance responses to non-self vocalizations while at the same time suppressing responses to self-generated sounds such as calls or respiration (*Shore and Zhou, 2006*). Multisensory integration of visual (*Bizley et al., 2007*) and olfactory information (*Cohen et al., 2011*) in the auditory cortex has been demonstrated in ferrets and rats, respectively. Evidence for multisensory integration in the primary auditory cortex also comes from studies on the modulation by touch (*Fu et al., 2003*; *Kayser et al., 2005*) and vision (*Kayser et al., 2007*) in primates.

## Modulation of responses to USVs by touch

The extensive modulation of primary auditory cortex by touch implies a possible role for touch-induced inhibition during social interactions where there is an increased vocalization. To test this, we analyzed the responsiveness of Au1 neurons to USVs in and out of touch. Interestingly, RS neurons exhibited a greater modulation to calls during touch as compared to out of touch. However, this was not the case with FS neurons. This large modulation of call responsiveness by touch could be construed as a sampling issue as calls during touch are fewer than those out of touch. However, bootstrapping analysis rules out this interpretation. Similarly, a higher intensity of stimulus calls during touch (due to close proximity) is less likely to be a contributing factor. Interestingly, there are several lines of evidence to suggest that inhibition in the auditory cortex could actually be responsible for an increased responsiveness to auditory stimuli. It has been reported that balanced inhibition underlies tuning and sharpens spike timing in auditory cortex (*Wehr and Zador, 2003*) and that inhibition might contribute to auditory processing (*Hamilton et al., 2013*; *Shamma, 2013*). In the light of these findings, it would be interesting to understand how multisensory response modulation of auditory cortex neurons affects the behavior of interacting animals.

## Materials and methods

### Animals

Wistar rats (45- to 60-day old, female and male) were commercially procured (Harlan, Eystrup, Germany) and housed with a 12:12 hr inverted light/dark cycle and *ad libitum* access to food and water. While implanted ('subject') rats were housed individually after surgery, 'stimulus' rats were housed in groups of 2–3. After a 1-week post shipment recovery, rats were handled for 2–3 days, following which they were habituated to the behavioral setup for 3–4 days. All experimental procedures were performed in accordance to German regulations on animal welfare (Permit no. G0259/09).

### Low-speed videography and behavioral analysis

The gap paradigm (*Wolfe et al., 2011*; *Bobrov et al., 2014*) was used to study social interactions between conspecifics, wherein facial touch episodes occur freely across a gap between rats placed on two platforms (30 cm × 25 cm, *Figure 1A*). The platforms (enclosed by 35 cm high walls on three sides) were elevated (20 cm) and placed in a Faraday cage to reduce electrical noise. The gap between the platforms was set to 20 ± 2 cm, depending on the size of the interacting partners. The entire setup was enclosed with black curtains and the room was darkened during experiments. An overhead low-speed camera (30 Hz) was used for continuous videography under infrared illumination (ABUS, Wetter, Germany). In most experiments, each recording session consisted of a 5 min baseline during which the subject rat was alone in the setup, a 5 min interaction time when stimulus animals/objects were presented across the gap, and another 5 min baseline at the end of interactions. On each recording day, 2–7 such recording sessions were conducted with various combinations of stimuli presented in a pseudo-random order. Stimuli presented included female and male conspecifics (60- to 120-day old), an object (Styrofoam block) and plastinated rats. These stimuli were presented either individually or in pairs (*Figure 1A*). Foam mats on the stimulus platform were changed between recording sessions to minimize olfactory cues. Offline analyses of videos were used to identify episodes of social facial interactions, and the following behavioral events were scored for: whisker overlap onset (*Figure 1B*, left), snout touch onset (*Figure 1B*, right), snout touch offset, and whisker overlap offset. The time of placement/removal of stimuli into/out of the setup was also scored for.

### Sound recording

Ultrasonic vocalizations produced by the rats were recorded using four microphones (condenser ultrasound CM16/CMPA, frequency range 10–200 kHz, Avisoft Bioacoustics, Berlin, Germany) placed under the elevated platforms (*Figure 1A*). Data were acquired using UltraSoundGate 416H at a sampling rate of 250 kHz and 16-bit resolution using Avisoft-RECORDER USGH software (Avisoft Bioacoustics, Berlin, Germany).

### Sound analysis

Acoustic analysis was performed using Avisoft SASLab Pro (Avisoft Bioacoustics, Berlin, Germany). Spectrograms were generated using the following fast Fourier transform (FFT) parameters: length of 1024 points and an overlap of 93.75% (FlatTop window, 100% frame size). The spectrograms had a frequency resolution of 244 Hz and a time resolution of 0.256 ms. Custom-written MATLAB codes (*Source code 1*) were also used to generate spectrograms using the Blackman–Harris window which resulted in a clearer appearance of frequency-modulated calls with steeper decays and a higher resolution (MathWorks, Natick, MA, USA). FFT parameters were: 500 points length and 80% overlap, resulting in 58 Hz frequency resolution and 0.4 ms time resolution. Start and end times of calls were manually set, and call category assignment was done as per an earlier classification (*Wright et al., 2010*). A total of 56,525 USVs were identified and classified by two different experimenters while being blind to the behavioral events. A small fraction of calls (5.8%) could not be unambiguously assigned into any one of these categories and were classified as 'unspecified'. In addition to call duration, mean frequency and band widths were computed. A subset (48,170 vocalizations) was acquired alongside recordings from the auditory cortex and used to compute neuronal responses to USVs.

To assign call source, low-speed videos of periods with only one animal present were superimposed with ultrasound recordings from all four channels. Analysis of these videos demonstrated a high degree of vocalization directionality. Hence, a comparison of call intensities on each of the channels was used to assign the source. A majority of USVs (80%) were assigned to either the subject or stimulus rats

while 20% remained unassigned (and subsequently omitted from all emitter-dependent analyses). To check if the fraction of unassigned calls was greater during touch episodes where close facial proximity could obscure the exact source, we analyzed 11,194 calls that occurred during touch. Only 17.8% of these were unassigned, which is similar to the overall unassigned fraction, ruling out any obfuscation of source assignment due to touch. Noise (generally not directional) or simultaneous calls (rare) perturb this algorithm in principle, but visual inspection of spectrograms showed that such interferences were the exception.

To estimate the accuracy of this method, we analyzed the false positive rates (i.e., calls wrongly assigned to stimuli) in two conditions: (i) recordings of subject rats interacting with plastinated rats and objects: in 13 recordings with 1833 calls, 1286 calls (70%) were correctly assigned to the subject, 284 calls (15%) were wrongly assigned to stimuli, and 263 calls (14%) were unassigned, (ii) recordings when subject rats were present alone on the setup: out of 4299 calls, 3406 calls (79%) were correctly assigned, 196 calls (5%) were wrongly assigned to stimuli, and 697 calls (16%) were unassigned. Having excluded the unassigned calls from emitter-dependant analyses, we hence estimate that >4 out of 5 calls were assigned to the correct source.

To compare intensity changes due to whisker touch, snout touch, and USVs, power spectral densities (PSD, as a function of time and frequency) were computed around these triggers. Integration over time verified that in the analyzed time window, power spectrum was indifferent to the trigger and frequency distribution was identical before and after it. Power was then cumulated over frequencies, which yields relative intensity as a function of time. Since the intensity values span several orders of magnitude, the geometric mean was used for comparison. Same number of triggers was randomly chosen to circumvent sample size issues. Interactions with vocalizations occurring in the time frame of interest as well as vocalizations that occurred simultaneously or in succession were excluded. All remaining episodes were averaged and traces plotted relative to the average during the pre-trigger phase. The pre-trigger average PSD was comparable across triggers (in the order of $10^{-8}$ V$^2$ Hz$^{-1}$).

## High-speed videography and whisker tracking

Whiskers of both subject and stimulus rats were tagged under 2–4% isoflurane anesthesia with small spherical drops of high-viscosity epoxy glue (Dymax 3021 UV-adhesive, Dymax Europe, Wiesbaden, Germany) that was hardened using ultraviolet light (Bluewave 50, Dymax Europe, Wiesbaden, Germany). The tag was covered with silver paint and fixed with superglue. Stimulus rats also received a black dot on the head to facilitate head tracking. Social interactions were recorded using a high-speed camera (A504k, Basler AG, Ahrensburg, Germany) at 250 Hz with 1280 × 1024 pixels. Video frames were streamed directly to a PCIe 1429 express card (National Instruments Corporation, Austin, TX, USA). Acquisition was controlled by custom-written Labview (National Instruments Corporation, Austin, TX, USA) programs. Whisker (one per animal) tracking was performed for a subset of 58 interactions (194 s of video) where the whiskers were clearly visible.

Tracking was done as has been described earlier (*Wolfe et al., 2011*; *Bobrov et al., 2014*) using a custom-written code (*Source code 2*). The process consisted of manual setting of head center and nose positions, automated contour detection and setting of whisker base and tag positions. Head axis (posterior to anterior) and whisker direction (outwards from whisker base) were acquired with custom-written MATLAB software. The head–whisker-angle was defined as zero in the orthogonal position, with protraction taking positive values up to a theoretical maximum of 180° and retraction being negative analogously. Whisker traces, that is, time series of whisker angle, were processed for signal cleansing, quality criteria, and signal decomposition as described elsewhere (*Hill et al., 2011*). This includes up-sampling to 1000 Hz and 1–25 Hz bandpass filtering (fourth order Butterworth filter). Whisker cycles were disregarded when not between 50 and 250 ms length or when amplitude was <7.5°. Hilbert transform yielded the phase within the whisking cycle starting at maximum protraction (0°), via maximum retraction (back-directed movement up to 180°) and return (≤360°). Whisking cycle phases at the onset of calls were analyzed for both the calling animal and the one towards which the calls were putatively directed.

To compute the locking of neuronal firing to whisker positions, spike time stamps for each cell were aligned to whisker traces and binned in a circular fashion before being normalized for phase occupancy. To the resulting polar plots, circular Rayleigh statistics were applied to calculate the Rayleigh vector (the summation of unit vectors in all individual angle observations to find a tendency in overall direction). The following criteria was set to identify putative phase locked cells: (i) minimum number of

spikes (10) in a tracked episode, (ii) Rayleigh vector greater than 0.2, and (iii) shuffling test (see below) showing significance at a confidence interval of 5%. The shuffling test is necessary because regular tests, such as Rayleigh or Hodges–Ajne test for non-uniformity of circular data, are sensitive to very low- or high-spike numbers because of the extreme (high or low) variance among bins. Also, the number of bins can affect the test result. To test independently of these, the spike train was randomly time shifted relative to the whisker phase trace. The Rayleigh vector was computed again and this procedure repeated 10,000 times to simulate a random set of Rayleigh vectors from the spike series. If the original Rayleigh vector was in the 95% quantile of that distribution, it was presumed to be phase locked.

## Analysis of whisking and calling periodicity

To analyze periodicity of calling, the intervals between successive calls were computed. For whisking cycles, the point of maximum protraction served as a marker. The spectral power density of whisking and calling was determined using Welch's method ('pwelch', MATLAB signal processing toolbox) in the range of 4–25 Hz. This was applied to the approximately continuous whisker trace and to the outline of the calling interval histogram, the latter with lower resolution and less power due to the rough discretization.

## Electrophysiology

Neuronal activity was recorded using a chronic microdrive (Harlan 8-drive, Neuralynx, Bozeman, MT, USA) consisting of eight independently movable tetrodes (arranged in a 4 × 2 array). The tetrodes were fashioned out of 12.5-μm diameter nichrome wire (California Fine Wire Company, Grover Beach, CA, USA) and gold-plated to a resistance of 250–300 kΩ (nanoZ, Neuralynx, Bozeman, MT, USA). The microdrives were implanted on 8 'subject' rats (four male, four female) spanning the following Bregma locations: −3.12 to −6.00 mm AP; 6.5 to 7.00 mm ML (*Paxinos and Watson, 2006*).

For implanting the drive, rats were subjected to ketamine (100 mg/kg body wt)/xylazine (7.5 mg/kg body wt) anesthesia. Booster doses of anaesthetics were administered as required. Body temperature was maintained with a heating pad and continuously monitored by a rectal probe (Stoelting, Wood Dale, IL, USA). After securing the animal's head onto a stereotactic apparatus (Narashige Scientific Instrument Lab., Tokyo, Japan), lidocaine was injected subcutaneously under the scalp. After retraction of the temporal muscle, the cleaned skull surface was treated with a UV-activated etchant-cum-glue (Optibond All-In-One, Kerr Italia, Salerno, Italy). Gold-plated screws were fixed away from the craniotomy site to anchor the drive and provide for grounding. After craniotomy and durectomy, the microdrive was positioned on the brain and the area was covered with 1% agarose. The microdrive was secured with dental cement (Paladur, Heraeus Kulzer, Hanau, Germany).

Tetrodes were lowered into the brain and recordings typically began 1–2 days after surgery. Tetrodes were advanced by a minimum of 80 μm between recording days to ensure that new high quality units were sampled during the course of the experiment. After passing through a unity-gain headstage, signals were transmitted via a tether cable to an amplifier (Digital Lynx, Neuralynx, Bozeman, MT, USA). Spike signals were amplified (10×), digitized at 32 kHz, and bandpass-filtered between 0.6 and 6 kHz. Events that crossed a user-set threshold were recorded for 1 ms (250 μs before and 750 μs after voltage peak).

At the end of the experiment, subject rats were deeply anaesthetized using ketamine/xylazine and electrolytic lesions (*Figure 3—figure supplement 1A*) were performed by injecting 10 μA negative current through the tetrodes for 10 s (nanoZ, Neuralynx, Bozeman, MT, USA). Transcardiac perfusion was performed with cold phosphate buffer and 4% paraformaldehyde. The brains were dissected out and post-fixed in 4% paraformaldehyde (overnight). Coronal sections (150 μm) were stained for cytochrome oxidase and visualized using light microscopy to identify the lesions. Recording depths (determined by number of microdrive turns) were used to identify exact recording sites and cortical layers (*Paxinos and Watson, 2006*) relative to lesions after accounting for shrinkage during tissue processing using Neurolucida (MBF Bioscience, Williston, VT, USA).

## Spike sorting and clustering

Amplitude and principal components were used for offline spike sorting using the semiautomatic clustering algorithm KlustaKwik (KD Harris, Rutgers University, Newark, NJ, USA). Manual correction and refinement were applied with MClust (AD Redish, University of Minnesota, Minneapolis, MN, USA)

using MATLAB (MathWorks, Natick, MA, USA). Spike features (energy and first derivative of energy) were used for separation. Inclusion criteria for single units were determined by refractory period, separation quality, and stability. Inter-spike interval histograms with minimal or no contamination in the first 2 ms were indicative of a single unit. Separation quality was determined by L-ratio (<0.2) and isolation distance (>15) (*Schmitzer-Torbert et al., 2005*). Stability of the units was quantified as follows: for each recording session, time periods outside interactions (as many in number and length as interactions on that day) were selected. This was performed on randomly distributed periods for 1000 permutations, and a linear correlation between time and firing rate was calculated. Average Pearson's R value was used as a measure of stability (higher R means stronger drift) and units with a value >0.4 were excluded from analysis.

## Cell classification

Spike shapes were used to classify units as putative regular-spiking (RS) or fast-spiking (FS) neurons. RS neurons have been shown to have wider action potentials while FS neurons on the other hand have narrower action potentials (*Atencio and Schreiner, 2008*). Auditory cortex single unit waveforms normalized by peak voltage were used to compute a whole host of features (height, trough, width parameters). Of these, two closely related features, full spike width and second half width showed bi-modal distributions (*Figure 3—figure supplement 1D,E*). For this analysis, the waveforms were compared to the widest spike in the dataset (which was designated the value of 1). Consequently, the spike widths of thinner spikes get assigned with negative values (*Figure 3—figure supplement 1D,E*). k-means clustering of units with two clusters resulted in well separated populations (indicated by dotted line, [*Figure 3—figure supplement 1F*]), albeit indicating a high correlation between these two features (arising possibly due to the spike waveforms being narrower than measured, which in turn is determined by filter settings). Indeed, spikes from these two populations resulted in well-defined average spike shapes (*Figure 3—figure supplement 1G*) and were well differentiated with respect to firing rates (*Figure 3—figure supplement 1H*).

## Response indices

To quantify the response of each neuron, we computed the average firing rate in the following response windows: in call duration alone and in a call duration + 25 ms window. For RS neurons, pair-wise comparison of basal firing rates were significantly different to the firing rates in the call duration alone (p = 0.0049) and call duration + 25 ms (p = 0.0006, Wilcoxon matched pairs signed rank test) response windows. However for FS neurons, the response was significantly different in the call duration + 25 ms window (p = 0.006), while it was not the case during the call duration alone (p = 0.0605). We also computed the onset responses in various time windows (0–25, 26–50, 51–75, 76–100 ms after call onset) as reported earlier (*Hromádka et al., 2008*). For RS neurons, we found significant increase in all four time windows while for FS it was significant only for the 26–50 and 51–75 ms time windows. It must be noted that since the stimuli are not all of the same duration (for e.g., trills: 50.9 ± 26.2 ms, mean ± SD, *Table 1*), the ideal response windows should span the entire call durations. Thus it appears that the call duration + 25 ms response window sufficiently well describes the responses of both RS and FS neurons in our study and this was used to compute the neuronal responses to USVs. For facial touch, firing rate was defined as average firing rate during all interactions with an interaction partner. The off-response due to facial touch was calculated as the mean firing rate in a 200 ms window after the end of the touch episode. These firing rates were compared with a matched baseline period, which was as long as the USVs/interactions and shifted −10,000 ms relative to these on the spike train (jumping over any periods which contained a USV or interaction). For each neuron, a response index was calculated as follows: Response Index = $(in - out)/(in + out)$, where *in* and *out* are the firing rates during a call/interaction and baseline firing rates, respectively.

## Statistics

Data were analyzed using Prism 6 (GraphPad Software Inc., La Jolla, CA, USA) or MATLAB (MathWorks, Natick, MA, USA) and are presented as mean ± SEM unless stated otherwise. Whisking phase preference of vocalizations was tested by Hodges–Ajne test. Since most of the data were not normally distributed (as determined by D'Agostino & Pearson omnibus normality test), differences between groups were tested with Wilcoxon signed rank test for paired data and Mann–Whitney U test for unpaired data. Comparison of distribution widths was performed by Kolmogorov–Smirnov test.

## Acknowledgements

This work was supported by Humboldt Universität zu Berlin, the Bernstein Center for Computational Neuroscience Berlin, the German Federal Ministry of Education and Research (BMBF, Förderkennzeichen 01GQ1001A), and Neurocure. EB was a recipient of a Berlin School of Mind & Brain scholarship. MB was a recipient of a European Research Council grant and the Gottfried Wilhelm Leibniz Prize. We thank Brigitte Geue, Undine Schneeweiß, Maik Kunert, and Juliane Steger for technical assistance; Viktor Bahr and Dominik Spicher for software development; Christian Ebbesen for sharing data; Carolin Mende for sharing art work; Ann Clemens, Christian Ebbesen, and members of the Brecht lab for valuable comments.

## Additional information

### Funding

| Funder | Grant reference number | Author |
| --- | --- | --- |
| Bernstein Center for Computational Neuroscience Berlin | | Michael Brecht |
| Bundesministerium für Bildung und Forschung | Forderkennzeichen 01GQ1001A | Michael Brecht |
| Charité Universitätsmedizin Berlin | Neurocure | Michael Brecht |
| European Research Council | | Michael Brecht |
| Deutsche Forschungsgemeinschaft | Gottfried Wilhelm Leibniz Prize | Michael Brecht |
| Humboldt Universität zu Berlin | Berlin School of Mind and Brain Scholarship | Evgeny Bobrov |

The funders had no role in study design, data collection and interpretation, or the decision to submit the work for publication.

### Author contributions

RPR, EB, Conception and design, Acquisition of data, Analysis and interpretation of data, Drafting or revising the article; FM, Analysis and interpretation of data, Drafting or revising the article; MB, Conception and design, Analysis and interpretation of data, Drafting or revising the article

### Ethics

Animal experimentation: All experimental procedures were performed in accordance to German regulations on animal welfare (Permit No. G0259/09).

## Additional files

### Supplementary files

• Source code 1. USV Classification. Custom-written MATLAB codes to generate spectrograms from multi-channel ultrasonic recordings. The codes also enable semi-automatic identification and manual classification of rat ultrasonic vocalisations.
• Source code 2. Whisker Tracking. Custom-written MATLAB codes to manually track rat whisker and head positions to compute whisking parameters (angle, phase etc).

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
