## [Decision Letter]

Thank you for sending your work entitled “Vocalization-whisking coordination and multisensory integration of social signals in rat auditory cortex” for consideration at *eLife.* Your article has been favorably evaluated by a Senior editor and 3 reviewers, one of whom is a member of our Board of Reviewing Editors.

The Reviewing editor and the other reviewers discussed their comments before we reached this decision, and the Reviewing editor has assembled the following comments to help you prepare a revised submission.

By analyzing responses in the auditory cortex of a rat interacting with another rat, the authors provide some interesting and novel evidence suggesting that whisker touch is coordinated with ultrasonic vocalization, and that responses in auditory cortex are modulated by touch. These are exciting results. However, the manuscript suffers from the fact that it is divided into two almost-independent parts, the first one dealing with the relation between whisking and vocalizations during facial interactions, and the second one describing the physiological correlates in auditory cortex. It would be desirable to further analyze the relationship between behavior and physiology, as described below. In addition, some of the major conclusions are not fully supported by the data that is presented.

Here is a summary of the major issues that need to be addressed in revision:

1) In the first part of the manuscript, the authors provide a detailed description of the type of calls rats emit during social interactions and the whisking behavior to which such calls are synchronized. However, this detailed information is not taken into account for the analysis of the physiological correlates. It would be interesting to know whether distinct types of calls can differentially modulate auditory cortex responses and whether there is a correlation between whisker position and modulation of firing rate.

2) The distinction between social and non-social touch is emphasized throughout the paper: “Surprisingly it appears that social but not non-social stimuli elicit this touch-related inhibition”. Yet the corresponding data is not shown. It is important that the data are presented as it is an unexpected result which will ultimately require further investigation. Furthermore, considering that during social touch other variables such as olfactory cues may be changing, this brings into question the conclusions of this paper that the effect is purely touch-mediated.

3) The authors should analyze neuronal responses to calls at higher temporal resolution (in and out of touch) rather than just integrating of longer temporal windows even extending beyond the end of the calls. Are there time locked on or off responses? How is the signal to noise ratio (e.g. Z-score) affected by touch? What is the effect of touch in a non-social setting? Bin size should be indicated.

4) The authors pooled recordings obtained from different sub-divisions of auditory cortex and recordings obtained from male and female animals. This may potentially introduce confounds. The authors should either limit their recordings to male or female and to A1 or AuV, or else, the data set should be extended to compare conditions.

5) The lack of responses by RS neurons could be accounted for by mismatch in frequency selectivity between the calls and the frequency responses of the neurons. For example, the Geffen lab (the Carruthers 2013 paper cited in the Introduction) used similar sounds and saw quite robust responses to vocalizations in rat A1. The simplest possibility is that the authors are recording in part of A1 that is not sensitive to ultrasonic frequencies; a change in 500 um can make a big difference in terms of the frequencies that neurons respond to, but it is difficult to tell from the coronal slices they show and the coordinates they describe, and they don't appear to do any kind of tone frequency mapping to figure out exactly where they are in A1. In addition, there could be a suboptimal choice of analysis windows (which were very long relative to the standard responses in auditory cortex). This issue can be resolved by more detailed analysis of the auditory responses.

6) The responses to touch in auditory cortex could be accounted for by sounds produced by the touch. The auditory system may be extremely sensitive to low-level broadband stimuli (see in the cat Bar-Yosef et al. 2007 and related papers). This could be resolved by (1) sensitive sound recordings (standard mics would probably not be sensitive enough for that); or (2) recordings from auditory cortex of acutely deafened rats.

7) The Discussion is very 'cortico-centric' – the auditory system has somatosensory-auditory interactions as early as the cochlear nucleus. This should also be considered in the Discussion.

---

## [Author Response]

We have provided new data, analyses and made revisions in the text and figures. Therefore, we believe that the revised manuscript is significantly strengthened after addressing key concerns expressed by the reviewers. Before we go on to provide a detailed response, we are listing below a summary of all of the major inclusions in the revised manuscript:

1) Data acquired from two more animals (our overall data set now stands at 4 males and 4 females)

2) Performed additional analysis to determine optimal response windows for modulation of auditory cortex neurons by ultrasonic vocalizations. We would particularly like to thank one of the reviewers for suggesting that we analyse earlier time windows. Indeed, we see several strong responses to calls and have reported this in the paper.

3) We also present detailed supplementary figures describing the heterogeneity of responses to calls seen in the auditory cortex, and responses to the different call categories.

4) We also present auditory cortex sub-region wise analyses, which have thrown up some interesting observations in relation to the effects of touch.

5) Extensive whisker tracking was also performed on of high-speed videos in order to analyse the relationship between whisking, calling and neuronal responses.

1) In the first part of the manuscript, the authors provide a detailed description of the type of calls rats emit during social interactions and the whisking behavior to which such calls are synchronized. However, this detailed information is not taken into account for the analysis of the physiological correlates. It would be interesting to know whether distinct types of calls can differentially modulate auditory cortex responses and whether there is a correlation between whisker position and modulation of firing rate.

The referee raises two important points here. First, the referee notes that we did not analyse the relationship between auditory cortex activity and whisking. We agree with the referee on the importance of this issue. In order to address this, we performed whisker tracking of high-speed videos and analysed the auditory cortex activity relative to the phase of whisking. As shown in Figure 3—figure supplement 5, we did not observe any systematic relationship between whisking phase and auditory cortex activity at the population level (in the 127 cells for which we obtained whisking data). A few cells did show a strong locking of firing rates with the retraction phase of the whisker, but this appeared be an artefact of the temporal coordination between whisking and calling. Indeed, these cells also showed a strong response to USVs.

Second, the referee asks about call type specific responses in our data set and we have now included a new supplementary figure with this information (Figure 3—figure supplement 2). In summary, RS neurons often show responses to more than one call type, whereas we do see examples of them being strongly modulated by a single call type. FS neurons on the other hand in general appear to less call selective, though examples to the contrary can also be seen. The panels in this supplementary figure also document the heterogeneity in responses to calls (early, late, sharp and sustained) that we observed.

Changes:

1) We performed whisker tracking along with auditory cortex recordings. The results thereof are shown in Figure 3—figure supplement 5.

2) Inclusion of Figure 3—figure supplement 2, which shows call-specific responses.

2) The distinction between social and non-social touch is emphasized throughout the paper: “Surprisingly it appears that social but not non-social stimuli elicit this touch-related inhibition”. Yet the corresponding data is not shown. It is important that the data are presented as it is an unexpected result which will ultimately require further investigation. Furthermore, considering that during social touch other variables such as olfactory cues may be changing, this brings into question the conclusions of this paper that the effect is purely touch-mediated.

In our earlier data set, we had observed a strong trend to this effect and had therefore made the claim. With the addition of new data, we however do not observe any significant modulation by social vs. non-social touch. We therefore retract the claim. While the effects of smell are important and certainly cannot be ruled out, the fact that both social and non-social touch induces inhibition suggests that the effect we observe is tactile in nature.

The statement has been deleted.

3) The authors should analyze neuronal responses to calls at higher temporal resolution (in and out of touch) rather than just integrating of longer temporal windows even extending beyond the end of the calls. Are there time locked on or off responses? How is the signal to noise ratio (e.g. Z-score) affected by touch? What is the effect of touch in a non-social setting? Bin size should be indicated.

First, the referee comments on the interesting idea of call responsiveness in and out of touch, given the strong effects of facial touch that we observe in the primary auditory cortex (Au1). Indeed in Figure 5, we report a much greater modulation of the neurons by USVs when in touch compared to out of it. Furthermore, this effect is seen in the regular-spiking but not fast-spiking neurons. Second, the referee asks about time locked on or off responses. We have included a supplementary figure (Figure 3—figure supplement 2) to demonstrate several example of time locked on responses. As far as off responses are concerned, we saw no clear examples, possibly due of the different call durations. One could however speculate that the sustained and delayed responses are possibly due to calls of various durations eliciting an off response in some cells. In our paradigm of freely interacting animals, the number of ‘trials’ (facial interactions) and ‘calls’ depends on the animals leading to very variable sampling episodes and consequently varying levels of significance. We think that for such a data set, our statistical assessment (which does not build on z-scores) is informative. Specifically, we show plots of raw spike rates, compute response indices and apply non-parametric tests to assess the significance of the responses. The referee also raises the issue of non-social touch, to which we have responded above.

Changes:

1) Analysis of neuronal response to calls in primary auditory cortex at higher temporal resolution is shown as earlier in Figure 5.

2) Representative examples of time locked responses are shown in the new supplementary figure (Figure 3—figure supplement 2).

3) The claim regarding differences between social and non-social touch has been deleted.

4) Bin sizes have been indicated in all the figure legends.

4) The authors pooled recordings obtained from different sub-divisions of auditory cortex and recordings obtained from male and female animals. This may potentially introduce confounds. The authors should either limit their recordings to male or female and to A1 or AuV, or else, the data set should be extended to compare conditions.

This suggestion by the referee indeed proved to be a great idea. Upon separating our dataset into the sub-regions of the auditory cortex, we observed the touch-evoked inhibition in Au1 but not in AuD or AuV (Figure 4—figure supplement 1). Therefore, we have restricted out analysis of neuronal responses to calls in and out of touch to Au1 (Figure 5). In addition, we have also expanded our analyses to show the responsiveness of neurons from these sub-regions to USVs (Figure 3—figure supplement 4). The suggestion to compare sex differences entails a large number of possible permutations and combinations (e.g. differences in calling behaviour and neuronal responses in relation to sex, estrous state, in and out of touch etc.). Inclusion of all this data is outside the scope of this manuscript and we would like to present this data in a separate manuscript.

Changes:

1) New supplementary figure (Figure 3—figure supplement 4) included showing sub-region wise differences in responsiveness to USVs.

2) New supplementary figure (Figure 4—figure supplement 1) included showing sub-region wise differences in responsiveness to touch.

3) Data in Figure 5 restricted to Au1 as touch effects are only observed here.

5) The lack of responses by RS neurons could be accounted for by mismatch in frequency selectivity between the calls and the frequency responses of the neurons. For example, the Geffen lab (the Carruthers 2013 paper cited in the Introduction) used similar sounds and saw quite robust responses to vocalizations in rat A1. The simplest possibility is that the authors are recording in part of A1 that is not sensitive to ultrasonic frequencies; a change in 500 um can make a big difference in terms of the frequencies that neurons respond to, but it is difficult to tell from the coronal slices they show and the coordinates they describe, and they don't appear to do any kind of tone frequency mapping to figure out exactly where they are in A1. In addition, there could be a suboptimal choice of analysis windows (which were very long relative to the standard responses in auditory cortex). This issue can be resolved by more detailed analysis of the auditory responses.

The referee raises the important issue of under-/sub-optimal sampling of the auditory cortex and states that this could be an issue with the lack of responses. We have now included new supplementary panels (Figure 3—figure supplement 1) to show the location of our recording sites over 8 animals and a range of Bregma values. Despite this, as the referee points out, we could still have been missing the USV responsive cells by being just slightly away. We did take up the suggestion of suboptimal choice of analysis windows and have indeed found much better responses to calls in shorter response windows. We would like to thank the reviewer for this particular suggestion. We analysed responses to calls in three shorter time windows: in the call duration alone, call duration + 25 ms and call duration + 50 ms. It must be noted here that since the stimuli are of varying durations (e.g. trills: 50.9 ± 26.2 ms, mean ± SD), the ideal response windows should span the entire call duration. We find strong responses in the call duration + 25 ms window for both RS and FS neurons and have used this to analyse the rest of the data (see Materials and Methods). In addition, we confirmed this fact by looking for onset responses in 0-25, 26-50, 51-75 and 76-100 ms windows as had been analysed earlier (22). We now not only show several examples of neurons activated strongly by USVs (Figure 3, Figure 3—figure supplement 2), but also report a significant modulation of the RS neurons by calls at the population level too (Figure 3).

Changes:

1) New supplementary panel (Figure 3—figure supplement 1) indicating the spread of the recording sites across the auditory cortex (in 4 females and 4 males).

2) New analyses to identify call duration + 25 ms window as the optimal response window.

3) Confirmation by use different of onset response time windows (0-25, 26-50, 51-75 and 76-100 ms).

4) Inclusion of panels to show examples of neurons activated strongly by USVs (Figure 3, Figure 3—figure supplement 2).

5) Reanalysis of modulation of the RS neurons by calls at the population level using the new response window (Figure 3).

6) The responses to touch in auditory cortex could be accounted for by sounds produced by the touch. The auditory system may be extremely sensitive to low-level broadband stimuli (see in the cat Bar-Yosef et al. 2007 and related papers). This could be resolved by (1) sensitive sound recordings (standard mics would probably not be sensitive enough for that); or (2) recordings from auditory cortex of acutely deafened rats.

The referee raises the possibility of the ‘sound of touch’ being responsible for the inhibition observed due to touch. We agree with the referee that this fact cannot be completely ruled out. To control for this in our paradigm, we measured the sound intensity levels in our recordings when triggered to the following events: onset of whisker-touch, onset of snout-touch and onset of USVs. While whisker-touch and snout-touch elicited little or no change of recorded audio power across all frequencies, USVs expectedly led to a large increase (Figure 4—figure supplement 1). Thus, at least according to the microphones, touch is not associated with increased sound pressure levels. We did not perform deafening experiments suggested by the referee because we currently lack the additional animal ethics permit required. We are also concerned that such a manipulation might alter patterns of multisensory integration and hence be less conclusive that anticipated.

Changes:

1) New supplementary panel showing the sound intensity levels in our recordings triggered to onset of whisker-touch, snout-touch and USVs (Figure 4—figure supplement 1).

2) Text has been added in the relevant Results and Discussion sections to state these results and the limitations of these analyses.

7) The Discussion is very 'cortico-centric' – the auditory system has somatosensory-auditory interactions as early as the cochlear nucleus. This should also be considered in the Discussion.

We agree that the previous manuscript had been cortico-centric. Yes, the role of cochlear nucleus deserves mention and we have included this. We thank the referee for pointing this out!

The relevant references have been cited in the Discussion.